# Enhanced Tensile Properties, Biostability, and Biocompatibility of Siloxane–Cross-Linked Polyurethane Containing Ordered Hard Segments for Durable Implant Application

**DOI:** 10.3390/molecules28062464

**Published:** 2023-03-08

**Authors:** Xiaofei Wu, Hanxiao Jia, Wenshuo Fu, Meng Li, Yitong Pan

**Affiliations:** College of Chemistry, Chemical Engineering and Materials Science, Shandong Normal University, Jinan 250014, China

**Keywords:** polyurethane, siloxane, ordered hard segments, tensile properties, biocompatibility

## Abstract

This work developed a series of siloxane-modified polyurethane (PU–Si) containing ordered hard segments by a facile method. The polyaddition between poly(ε-caprolactone) and excess diurethane diisocyanate was carried out to synthesize a polyurethane prepolymer with terminal isocyanate groups, which was then end-capped by 3-aminopropyl triethoxysilane to produce alkoxysilane-terminated polyurethane; the target products of PU–Si were obtained with hydrolysis and the condensation of alkoxysilane groups. The chemical structures were confirmed by FT-IR and XPS, and the effect of the siloxane content or cross-linked degree on the physicochemical properties of the PU–Si films was investigated in detail. The formation of the network structure linked by Si–O–Si bonds and interchain denser hydrogen bonds endowed PU–Si films with fine phase compatibility, low crystallinity, high thermal stability, and excellent tensile properties. Due to the high cross-linked degree and low interfacial energy, the films displayed a high surface water contact angle and low equilibrium water absorption, which resulted in slow hydrolytic degradation rates. Furthermore, the evaluation of protein adsorption and platelet adhesion on the PU–Si film surface presented high resistance to biofouling, indicating superior surface biocompatibility. Consequently, the siloxane–cross-linked polyurethane, which possessed excellent tensile properties, high biostability, and superior biocompatibility, showed great potential to be explored as biomaterials for durable implants.

## 1. Introduction

Polyurethane (PU), a multifunctional synthetic material, can be applied in many fields of foams, coatings, adhesives, sealants, packaging, and construction materials due to its toughness, flexibility, durability, and anti-abrasion resistance [1,2]. The distinct polarity of hard and soft segments produces microphase separation, and the hard domains serve as fillers dispersed in soft substrates and generate enhanced tensile properties to the PU materials. It is worth mentioning that the tensile properties are largely driven by the interchain hydrogen bonds formed among the hard segments. Moreover, the two-phase microstructures are similar to the cytomembrane structures, which endow PUs with good biocompatibility. By varying the species and composition of soft and hard segments, PUs with adjustable tensile properties can be used as biomaterials for the regeneration of rigid and soft tissues, such as cartilages, knee-joint meniscus, catheters, and prosthetic skins [3,4,5].

However, when PUs are used as durable implants, the adverse reaction of surface biofouling can damage the tensile properties and reduce biostability, which shortens their service life and even leads to the failure of implanted devices [6]. To repel biofouling and build a bioinert surface, many strategies have been developed to produce a polishing surface. Among them, coating and surface grafting are the popular approaches to achieving high surface biocompatibility [7,8,9]. Nevertheless, coatings adhering to the surface mainly through noncovalent interactions tend to peel off from the surface during the application process [10]. Although surface grafting can overcome the drawback of coating, the techniques, including plasma activation, strong oxidation, UV irradiation, and chemical treatment, are usually complicated and inevitably deteriorate the native properties of substrates [11,12,13]. The Hou group prepared a category of medical PUs with phosphorylcholine (PC) on the terminal of the side chain by bulk modification [14]. The PC groups possessing high mobility gathered on the surface after the material was in contact with water and produced a surface with high anti-biofouling capacity. Furthermore, the cast films exhibited excellent tensile strength (>40.8 MPa) and stretchability (>992%). For the durable implant application, it is also necessary to develop effective strategies to prepare medical PUs maintaining superior tensile properties, high biostability, and surface biocompatibility.

Polysiloxane is universally utilized for the manufacture of biomedical devices due to its high flexibility, low toxicity, and superior biostability [15,16,17]. Moreover, its low surface energy can produce good hydrophobicity and reinforce the resistance to biofouling [18]. Siloxane-based PUs combine the merits of the properties of silicone and PU, and several strategies have been developed to prepare siloxane-modified PUs [19,20,21]. Among these, the incorporation of siloxane units into the final PU as soft segments is the most useful technique. However, excessive microphase separation is often found because of high incompatibility between nonpolar siloxane and polar PU; this kind of polymer exhibits relatively poor tensile properties in most cases [22]. To overcome this drawback, alkoxysilane, which can produce hydrolysis and condensation under moisture, is used to modify PU. Recently, Teng et al. reported a PU with surface-grafting alkoxysilane that self-condensed in the existence of water to produce a covalent-bonding coating on the surface [23]. The resulting polymers possessed superior tensile properties and surface anti-biofouling capacity compared with parent PU. Although many strategies have been developed to prepare siloxane-based PU elastomers and foams with special functions, such as high elasticity, shape memory property, thermal-insulating properties, water/oil separation, and improved biocompatibility [24,25,26,27], the research on siloxane-modified PU materials is still worth investigating.

In this study, a series of siloxane–cross-linked PU containing ordered hard segments (PU–Si) was designed and prepared by a facile method. They were expected to own enhanced tensile properties, biostability, and biocompatibility for the application of durable implants. The polyaddition was first carried out between poly(ε-caprolactone) (PCL) and diurethane diisocyanate (HBH) to synthesize a prepolymer with terminal isocyanate groups (PUP–NCO), which was then end-capped by 3-aminopropyl triethoxysilane (APTS) to produce alkoxysilane-terminated polymer (PUP–SiOR); finally, the target products of PU–Si were obtained with the self-cross-linking of alkoxysilane groups under moisture. The effect of siloxane-based cross-linking on the structural and physicochemical performances of the PU–Si films was investigated. The surface anti-biofouling capacities (biocompatibility) were determined with the methods of protein adsorption and platelet adhesion.

## 2. Results and Discussion

### 2.1. Synthesis

It is known that PUs prepared with aromatic isocyanate have lower biocompatibility than those based on aliphatic isocyanates because aromatic diamines, the degradation products from the hard segments, were reported to be carcinogenic [28], although the physiological concentration of the products remained controversial [29]. However, the PUs based on aliphatic isocyanates, usually possessing poor tensile properties, were not suitable for durable implant application. The triblock aliphatic diisocyanate (HBH) with two urethane groups in its structure was successfully synthesized through an ordinary condensation between 1,4-butanediol and excess hexamethylene diisocyanate. Due to the denser hydrogen bonds among hard segments, the prepared PU based on HBH exhibited excellent tensile properties [30]. In the work, PUP–NCO was obtained through the reaction of PCL with an excess amount of HBH, and the –NCO content in the system determined with di-n-butylamine titration was used to monitor the progress of the reaction. After the –NCO groups reacted with –NH_2_ of APTS, the alkoxysilane groups were introduced to the chain end. The reaction endpoint was confirmed through the vanishment of the –NCO absorption in FT-IR. It is worth noting that the reaction should be performed at low temperatures because of the much higher activity of –NH_2_ than –OH. The content of alkoxysilane groups was easily controlled by varying the mole ratio of BHB/PCL. Due to the moisture sensitivity, a network structure linked by Si–O–Si bonds was produced through hydrolysis and the condensation of alkoxysilane groups under moisture. Since there were three alkoxysilane groups at each end, only a small amount of APTS could reach the target of cross-linking modification. In the meantime, the film materials were obtained after solvent evaporation.

### 2.2. FT-IR Analysis

The FT-IR spectra of PCL, HBH, APTS, and PU–Si film (taking PU–Si–III as a representative sample) are shown in Figure 1. The FT-IR spectrum of PCL (Figure 1a) exhibited strong peaks at 1720 and 1166 cm^−1^, which were assigned to the absorption of aliphatic esters C=O and C–O–C, respectively. The stretching vibration of -CH_2_- was observed at 2940 and 2862 cm^−1^. The weak absorption band at ~ 3513 cm^−1^ belonged to the terminal –OH, which vanished after the reaction with excess HBH. In the HBH spectrum (Figure 1b), the typical absorption peak of free –NCO groups was presented at 2268 cm^−1^. In addition, the observed peaks of –NH– (3312 cm^−1^), amide I (1673 cm^−1^), and amide II (1530 cm^−1^) indicated the existence of urethane groups in HBH. The absence of –NCO and –NH_2_ (1410 cm^−1^, Figure 1c) absorption peaks in the PU–Si–III spectrum (Figure 1d) demonstrated the complete reaction between PUR–NCO and APTS, and no residual APTS in the product. In addition to the absorption peaks of urethane and ester groups, an absorption band with high intensity present at 1044 cm^−1^ belonged to the characteristic Si–O–Si stretching vibration [31], which confirmed the network structure linked through Si–O–Si bonds.

### 2.3. XPS Analysis

The surface elemental content of PU–Si films was measured with XPS, and the overview and narrow charts are presented in Figure 2. From the overview XPS charts (Figure 2a), it was found that the characteristic signals of carbon (C1s), nitrogen (N1s), and oxygen (O1s) atoms arose at 529.9, 400.7, and 285.6 eV, respectively. The presence of new signals at 152.9 and 101.7 eV was respectively assigned to Si2s and Si2p [32], which manifested that the silicon element was successfully introduced to the films (PU–Si–II~–V). In addition, the intensity for Si2s and Si2p signals presented a steady increase (Figure 2b), which was in accordance with the silicon element content in the film samples. The XPS analysis provided indirect evidence for the preparation of PU–Si.

### 2.4. XRD Analysis

XRD was commonly used to analyze the crystallization behavior of polymer materials, and the diffractograms of PCL and PU–Si films are presented in Figure 3. All the PU–Si films showed only two blunt diffraction bands at about 21.3° and 24.3°, which corresponded to the crystalline of the PCL soft segments [33] and ordered HBH hard segments [30], respectively. It indicated that there existed crystalline regions in these films. The intensity of the diffraction bands lessened as the siloxane content increased (PU–Si–I~–V), which meant that the crystallinity of the PU–Si films progressively weakened. Especially for the PU–Si–V film that contained the highest siloxane content in the samples, the weak and broad peak implied an amorphous structure. This phenomenon was caused by the Si–O–Si cross-linked network, which deterred the orderly arrangement of the chain segments, generating a decrease in crystallinity [34].

### 2.5. DSC Analysis

The DSC curves and corresponding data for PU–Si films are shown in Figure 4 and Table 1, respectively. All the samples displayed similar thermal transition with glass transition temperatures (*T*_g_) at 26.3~12.2 °C and crystallization temperature (*T*_c_) at 76.8~42.9 °C. Only one *T*_g_ in each curve indicated that there was no obvious microphase separation [35], and the films possessed good compatibility between the phases of PU and siloxane. With the increase of siloxane content in the samples, the values of *T*_g_ and *T*_c_ decreased gradually, which were ascribed to the high freedom degree of Si–O–Si bonds [36]. Clearly, high alkoxysilane content endowed the films with high cross-linked density, which reduced molecular chain movement and weakened the crystallization ability, resulting in reduced crystallinity and fusion enthalpy (Δ*H*_m_) (Table 1) [31]. Such low *T*_g_ and Δ*H*_m_ meant that the PU–Si films were in an elastomer state at room temperature and could be used in a wide temperature range.

### 2.6. TGA Analysis

The TGA and DTGA curves of PU–Si films are depicted in Figure 5, and the relevant data are summarized in Table 2. The films exhibited a decomposition temperature at 5% weight loss (T_5%_) higher than 200 °C, manifesting high thermal stability. Two degradation stages in the weight loss process were found for all the samples: the weight loss at the first stage with a maximum decomposition temperature (T_max–1_) of 263~334 °C was primarily the dissociation of urethane (C–N: 305 kJ/mol^−1^) and ester bonds (C–O: 326 kJ/mol^−1^); in the second stage, the slight weight loss with the T_max–2_ above 400 °C belonged to the dissociation of Si–O bonds (460 kJ/mol) [37]. In addition, the T_max_ values increased with the increase of alkoxysilane content. The enhanced thermal stability was ascribed to the Si–O–Si cross-linked structure, which produced a strong interaction among molecular chains and improved the thermal stability. Obviously, the residue at the test endpoint was mainly SiO_2_, and the values of residual weight (W_r_) were consistent with the alkoxysilane content in the films.

### 2.7. Tensile Properties

The tensile properties of the PU–Si films were acquired using their derived stress–strain curves (Figure 6), and the summary data, including ultimate tensile strength (UTS), ultimate elongation (UE), Yield modulus (YM), and fracture toughness (FT), are presented in Table 3. All the samples exhibited a yield point, meaning a transformation from elasticity to plasticity. With the increase of alkoxysilane content, the UTS and YM increased markedly, while the UE showed a reverse trend, which was mainly attributed to the increasing cross-linking degree in the films [38]. The Si–O–Si cross-linked structure endowed the films with excellent tensile properties. For example, PU–Si–III, which had a medium cross-linking degree in the film samples, displayed a UTS of 38.4 MPa, UE of 797%, YM of 50.6 MPa, and FT of 21.6 MJ/m^3^. Due to the multiple urethane groups in each ordered hard segment, denser hydrogen bonds were formed between the molecular chains, which was beneficial for improving their tensile properties. The UTS and UE of PU–Si films compared favorably with those of the MDI–based PU in biomedical applications, such as Biospan^TM^ (UTS: 46 MPa; UE: 680%) and Chronoflex AR^TM^ (UTS: 41 MPa; UE: 450%) [39]; the results indicated that the tensile properties met the requirement for durable implants.

### 2.8. Surface and Bulk Hydrophilicity

The surface water contact angle (WCA) and water absorption (WA) were adopted to characterize the surface and bulk hydrophilicity of the PU–Si films, respectively. From the WCA data shown in Figure 7a, it was found that, as the alkoxysilane ratio increased (PU–Si–I~–V), the WCA values increased from 80.3° to 115.3°, manifesting an obvious decline in surface wettability. The improvement of surface hydrophobicity was ascribed to the low-polarity siloxane on the film surface, which caused the reduction of surface free energy [40,41]. Generally, high surface hydrophobicity meant superior anti-biofouling capacity; thus, it was speculated that the PU–Si materials possessed better surface biocompatibility than that of the PUs.

It is well known that bulk hydrophilic ability has a crucial effect on the hydrolytic degradation rate, which is closely related to the biostability of implant devices [42]. The plotted diagrams of time-coursed WA for PU–Si–films are shown in Figure 7b. The WA increased rapidly at the preliminary stage and achieved equilibrium after 48 h. All the films exhibited low equilibrium water absorption (EWA), and the EWA values steadily decreased from 8.4% to 2.5% with the increase of alkoxysilane ratio in the films (PU–Si–I~–V). This phenomenon was elucidated by the incremental surface hydrophobicity and cross-linking degree. The surface hydrophobicity produced by siloxane segments in the polymer structures improved their water resistance and hindered water molecules from diffusing through the film. The formation of a cross-linked structure with covalent Si–O–Si bonds and noncovalent hydrogen bonds produced a denser network and effectively reduced the EWA.

### 2.9. In Vitro Degradation

The in vitro degradation behaviors for the PU–Si film in PBS at 37 °C were plotted and are shown in Figure 8. All the film samples showed a mass loss of less than 10% after degradation for 8 months and maintained their integrity after degradation for 20 months, indicating a slow degradation rate or high biostability. With the incremental siloxane content and cross-linked degree in the films, an obvious reduction in the degradation rate was found, which corresponded to the trend of EWA. For example, PU–Si–I, a linear chain polymer, displayed 42.5% mass loss after degradation for 20 months, and it was deduced that the sample basically lost its tensile properties. Conversely, only 11.4% mass loss for PU–Si–V, which had the maximum siloxane content and cross-linked degree, was found at the end of the measurement, and the sample seemed to retain most of its tensile properties. The degradation primarily occurred through the hydrolysis of ester bonds in PCL segments; thus, high EWA deterred the water molecules from reaching the ester bonds, generating a reduced degradation rate.

The degradation process was intuitively reflected through the surface morphologies. Figure 8b shows the SEM images of PU–Si–III film at predetermined degradation times. A smooth surface for the pristine film was observed (Figure 8b(i)). As the degradation proceeded, the surface became rougher and rougher (Figure 8b(ii,iii)), while only a slight mass was lost. After degradation for 20 months, many irregular holes appearing on the surface signified that some degradation products were dissolved in PBS (Figure 8b(iv)).

### 2.10. Surface Biocompatibility

The biocompatibility of the PU–Si films was assessed through the anti-biofouling ability on the surface, which was performed by protein adsorption and platelet adhesion. Figure 9a displays the adsorbed BSA on the PU–Si film surface. The unmodified sample of PU–Si–I presented a large amount of absorbed BSA (13.5 μg/cm^2^), meaning a relatively low ability for protein resistance. As expected, a higher protein–resistant surface was acquired after the incorporation of siloxane into the materials, and the adsorbed BSA was reduced from 8.3 μg/cm^2^ to 2.8 μg/cm^2^ with the increase of siloxane content. The Si–O–Si cross-linked network endowed the films with low interfacial energy, which effectively reduced the interactions between protein molecules and surface, producing an improvement of protein-resistant characteristics. The same phenomenon was found in siloxane–polyurethane coatings [43] and PDMS-modified polyurethane [44]. The platelet adhesion on the PU–Si film surfaces, as shown in Figure 9b, exhibited a similar trend with protein adsorption. The number of platelets adhered to the surface presented a substantial reduction as the siloxane content increased (PU–Si–I~–V), which was also attributed to the reduced interfacial energy. Furthermore, the adhered platelets maintained their pristine states, and no deformation and aggregation were found, indicating a weak interaction between platelets and the surface. All the results demonstrated that the PU–Si materials possessed superior biocompatibility and could be explored as anti-biofouling materials for biomedical applications.

## 3. Materials and Methods

### 3.1. Materials

APTS (98.5%), dibutyltindilaurate (DBTDL, 95%), and PCL (*M*_n_: 1950 g/mol) were J&K reagents. *N,N*-dimethylformamide (DMF) with moisture content less than 0.1% was obtained from Macklin (Jinan, China). HBH was synthesized according to the reported method [10]. Other chemicals were of AR grade.

### 3.2. Preparation of PU–Si and PU–Si Films

The codes and basic compositions of PU–Si films were shown in Table 4. The experimental process of PU–Si–V was as follows: PCL (4.0 mmol) and HBH (8.0 mmol) were dissolved in DMF (0.25 g/mL). Under the protection of dry N_2_, a drop of catalyst was added, and the reaction was performed at 80 °C to obtain the PUP–NCO (~1.5 h). The conversion of the PUP–NCO into the PUP–SiOR was carried out by adding a DMF solution of APTS (4.0 mmol, 0.25 g/mL) at 20~25 °C by drops under mechanical stirring. After the disappearance of –NCO absorption in FT-IR, additional DMF was introduced to the system (~0.04 g/mL). The diluted solution was first placed under reduced pressure to remove the bubbles and then cast into a Teflon mold. The uniform PU–Si film (thickness: 0.28 ± 0.2 mm) was obtained by curing in the air with 50% relative humidity and solvent evaporation at 45 °C. The residual DMF was thoroughly removed by vacuum drying. Other samples were prepared using the same procedure, according to the recipes in Table 4. The preparation route for PU–Si is illustrated in Figure 10.

### 3.3. Characterization

FT-IR measurements were performed with a NICOLET 6700 infrared spectrometer (Thermo–Scientific, Waltham, MA, USA) in the region of 4000–400 cm^−1^ at 4 cm^−1^ resolution. X-ray photoelectron spectroscopy (XPS) tests were conducted using a Thermo ESCALAB250Xi spectrometer (Thermo–Scientific, Waltham, MA, USA) with a monochromatic Al–Kα radiation source. Thermogravimetric analysis (TGA) was acquired using a TGA 2050 analyzer (Universal, New Brunswick, NJ, USA) with a ramp of 20 °C/min under N_2_. Differential scanning calorimetry (DSC) was performed with a DSC2500 (TA Instruments, New Castle, DE, USA) with a temperature range of −60~150 °C, and the curves were obtained from the second heating process at a heating ramp of 10 °C/min. The patterns of X-ray diffraction (XRD) were collected with a D8 AVANCE instrument (Bruker, Rheinstetten, Germany) with the diffraction angle 2θ of 5~55°. Tensile testing was performed with an LDS–05 testing machine (Liling Instruments, Jinan, China) according to the ISO527–2 standard. Dumbbell-shaped samples with a gauge width of 5.0 mm were tested using an elongation rate of 25 mm/min. The surface static water–air contact angle was tested with a KSV CAM 200 (Helsinki, Finland) contact angle meter. Bulk hydrophilicity of the films was quantified through water absorption using a gravimetric increment method in deionized water at 37 °C. In vitro degradation was executed in phosphate buffer saline (PBS, pH 7.4) at body temperature over 20 months, and the percentages of mass loss at determined time intervals were used to analyze the degradation rate. The degraded samples were collected, freeze-dried, and observed with a scanning electron microscope (SEM, Quanta 200, FEI, Eindhoven, The Netherlands). The surface biocompatibility (anti-biofouling capacity) was evaluated with protein adsorption in bovine serum albumin (BSA) and platelet adhesion in platelet-rich plasma (PRP), according to the reported methods [45].

## 4. Conclusions

This work developed a kind of siloxane–cross-linked polyurethane (PU–Si) containing ordered hard segments for medical applications. The influence of the siloxane content and cross-linked degree on the physicochemical properties of PU–Si films was investigated in detail. The network structure linked by Si–O–Si bonds and interchain denser hydrogen bonds endowed the PU–Si films with fine phase compatibility, low crystallinity, high thermal stability, and excellent tensile properties. The high surface WCA contact angle and low EWA meant high surface and bulk hydrophobicity. The results of in vitro hydrolytic degradation showed that PU–Si films had high biostability, with a mass loss of less than 10% after degradation for 8 months. Furthermore, because of the low interfacial energy, the PU–Si films presented high resistance to biofouling (protein adsorption and platelet adhesion), manifesting superior biocompatibility. Consequently, the PU–Si films, which simultaneously possessed excellent tensile properties, high biostability, and superior biocompatibility, could be explored as biomaterials for durable implant application.

## Figures and Tables

**Figure 1 molecules-28-02464-f001:**
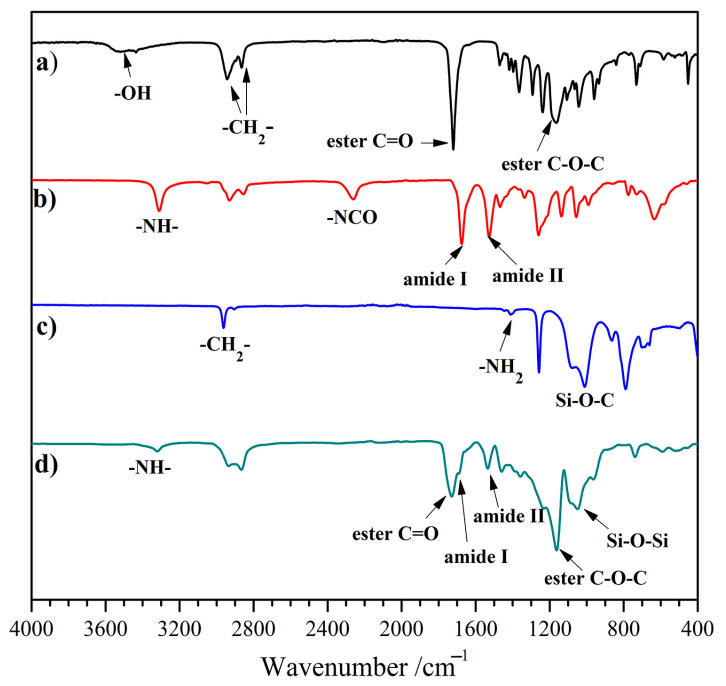
FT-IR spectra of (**a**) PCL, (**b**) HBH, (**c**) APTS, and (**d**) PU–Si–III.

**Figure 2 molecules-28-02464-f002:**
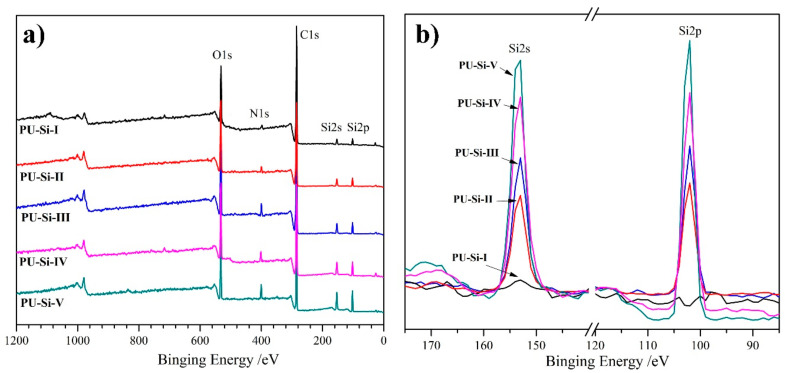
(**a**) Overview and (**b**) narrow XPS charts of PU–Si films.

**Figure 3 molecules-28-02464-f003:**
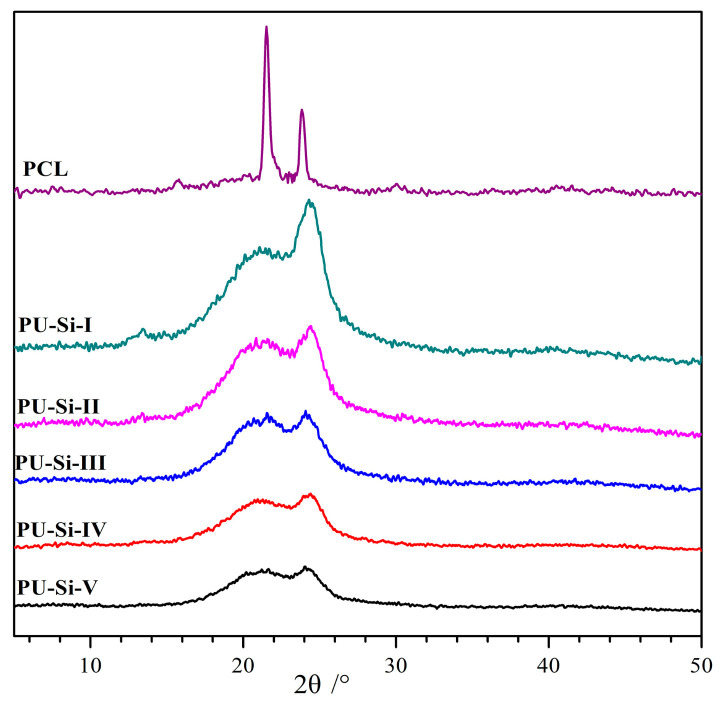
XRD diffractograms of PCL and PU–Si films.

**Figure 4 molecules-28-02464-f004:**
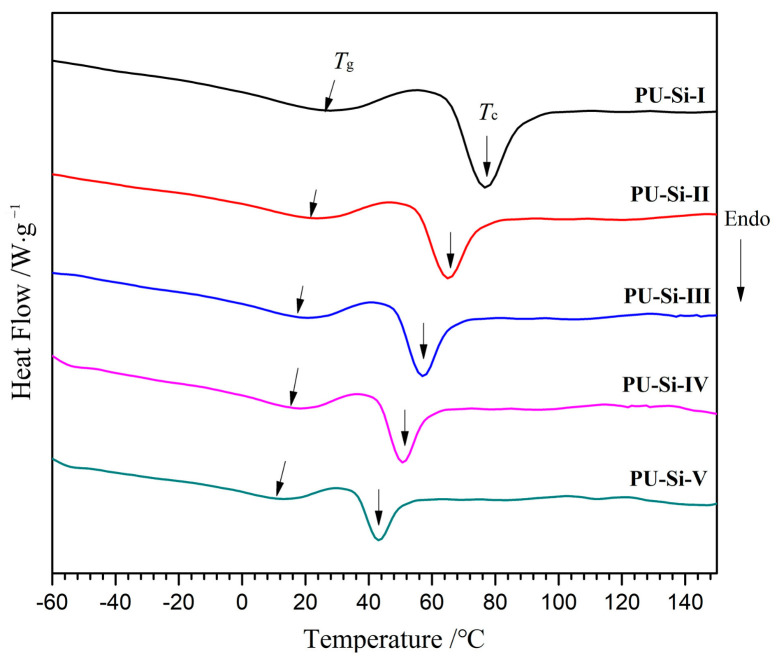
DSC curves of PU–Si films.

**Figure 5 molecules-28-02464-f005:**
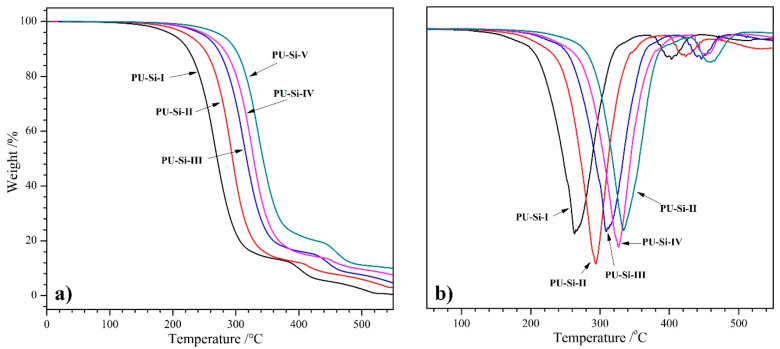
(**a**) TGA and (**b**) DTGA curves of PU–Si films.

**Figure 6 molecules-28-02464-f006:**
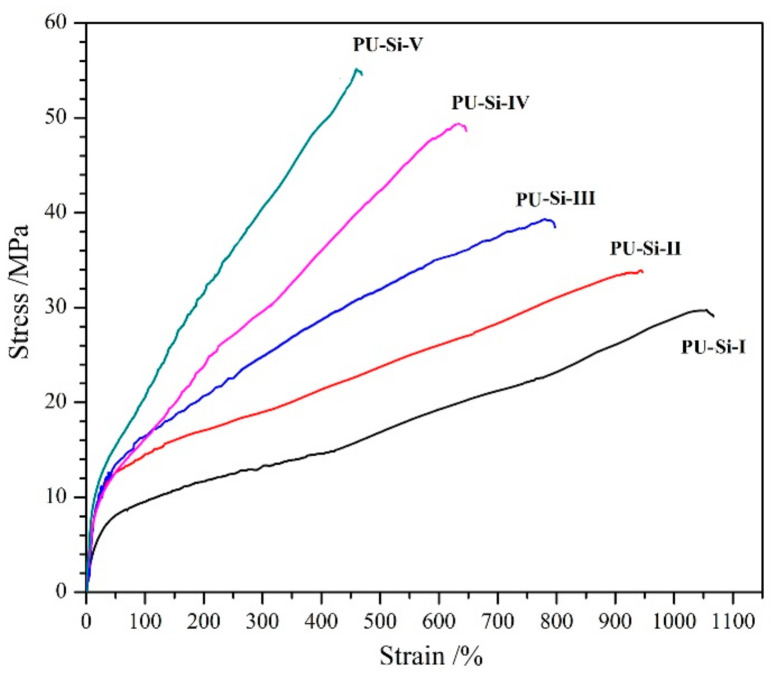
Typical stress–strain curves of PU–Si films.

**Figure 7 molecules-28-02464-f007:**
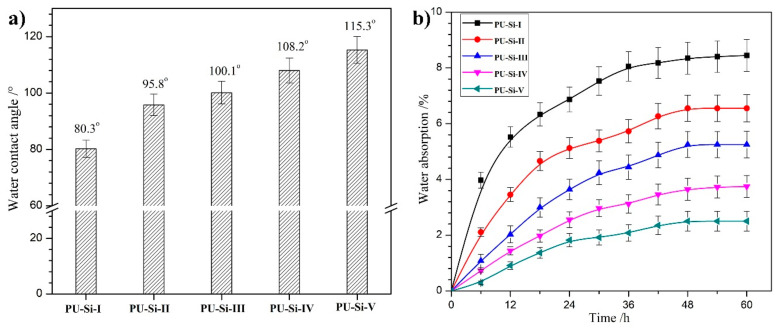
(**a**) Surface contact angle (*n* = 5) and (**b**) water absorption (*n* = 3) of PU–Si films.

**Figure 8 molecules-28-02464-f008:**
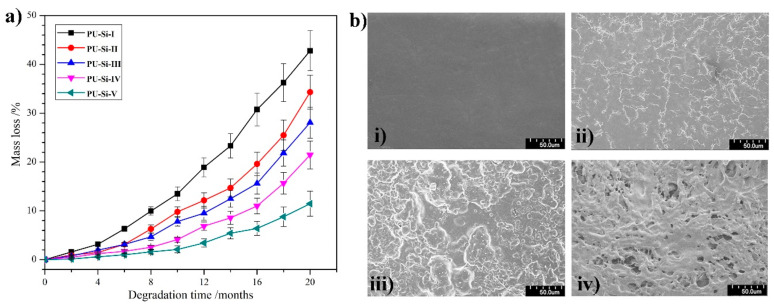
(**a**) In vitro degradation behaviors of PU–Si films (*n* = 3); (**b**) typical SEM images of PU–Si–III film after degradation for (i) 0, (ii) 8, (iii) 16, and (iv) 20 months.

**Figure 9 molecules-28-02464-f009:**
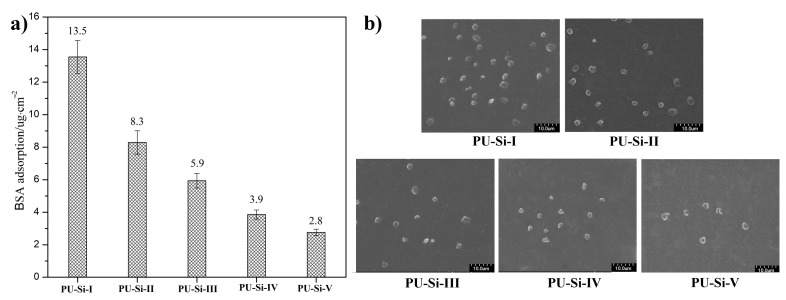
(**a**) BSA adsorption (*n* = 3) and (**b**) platelet adhesion on the PU–Si film surface.

**Figure 10 molecules-28-02464-f010:**
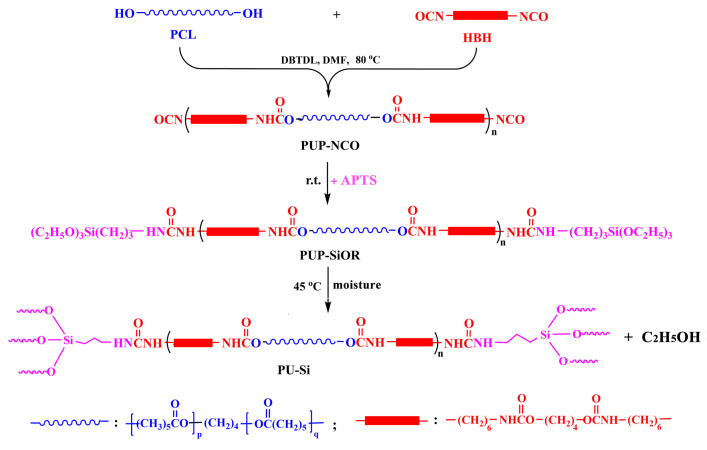
The preparation route of PU–Si.

**Table 1 molecules-28-02464-t001:** Data of PU–Si films from DSC thermograms.

Film Samples	*T*_g_/°C	*T*_c_/°C	Δ*H*_m_/J·g^−1^
PU–Si–I	26.3	76.8	19.1
PU–Si–II	22.0	65.3	15.7
PU–Si–III	17.8	57.0	13.6
PU–Si–IV	15.1	50.8	11.2
PU–Si–V	12.2	42.9	8.9

**Table 2 molecules-28-02464-t002:** Data of PEU–Si films from TGA and DTGA curves.

Film Samples	T_5%_/°C	T_max–1_/°C	T_max–2_/°C	W_r_/%
PU–Si–I	201	263	404	0.8
PU–Si–II	226	293	423	3.1
PU–Si–III	248	309	446	4.9
PU–Si–IV	255	326	454	7.5
PU–Si–V	281	334	460	10.1

**Table 3 molecules-28-02464-t003:** Tensile properties of PU–Si films (n = 3).

Film Samples	UTS/MPa	UE/%	YM/MPa	FT/MJ·m^−3^
PU–Si–I	29.8 ± 1.9	1064 ± 85	23.2	19.1
PU–Si–II	33.7 ± 2.2	944 ± 68	41.7	21.2
PU–Si–III	38.4 ± 2.7	797 ± 53	50.6	21.6
PU–Si–IV	49.2 ± 3.3	633 ± 52	57.2	19.7
PU–Si–V	54.7 ± 3.8	475 ± 31	70.8	16.9

**Table 4 molecules-28-02464-t004:** Codes and compositions of the PU–Si films.

Film samples	PCL/mmol	HBH/mmol	APTS/mmol	HBH /wt%	APTS/wt%
PU–Si–I	4.0	4.0	0	17.5	0
PU–Si–II	4.0	5.0	2.0	20.2	4.2
PU–Si–III	4.0	6.0	4.0	22.4	7.7
PU–Si–IV	4.0	7.0	6.0	24.2	10.8
PU–Si–V	4.0	8.0	4.0	25.9	13.4

## Data Availability

Not applicable.

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
