# Peer review of "Enhanced Tensile Properties, Biostability, and Biocompatibility of Siloxane–Cross-Linked Polyurethane Containing Ordered Hard Segments for Durable Implant Application"

_molecules, 2023, doi:10.3390/molecules28062464_

Round 1

Reviewer 1 Report

This is an interesting manuscript. However, I do not agree that "the research on siloxane-modified PU materials remains relatively limited" and I recommend to add more examples of modification of polyurethanes (including PU foams) properties with functional silanes and polysiloxanes of different chemical structures, which have been described in a scientific literature.

Moreover, this manuscript requires some other corrections.

1. In lines 76 - reaction of poly(ε-caprolactone) terminated with hydroxyl groups (PCL) with diurethane diisocyanate is polyaddition, but not polycondensation !

2. What was a chemical structutre of diurethane diisocyanate (HBH) ?

3. I also sugest to present more detailed description of FTIR spectra, containing kinds of absorption bands corresponding to appropriate chemical bonds.

4. I suggest to use more common abbreviation for 3-aminopropyl-(triethoxy)silane, e.g., APTS or APTES.

5. Many other symbols should be explained: XPS, TGA, TG, XRD, PBS, DSC, and SEM.

6. In line 296 I propose to change a word "introduction" for "incorporation".

Reviewer 2 Report

The paper entitled “Enhanced Tensile Properties, Biostability and Biocompatibility of Siloxane-Crosslinked Polyurethane Containing Ordered Hard Segments for Durable Implant Application” focus on series of siloxane-crosslinked PU containing ordered hard segments. The authors have shown that the unique properties of polysiloxane, such as low surface energy, good thermal stability, and excellent flexibility, are mainly attributed to its intrinsic structure contained inorganic Si–O bonds.

Comments:

1. Figure 1. Since in the third stage of the reaction process water must react and ethyl alcohol should be released, this must be reflected in the reaction scheme. What does 50% RH mean in the third stage of the process. Explanation of the structure of RH is not found in the text.

2. Figure 4. At high angles, this type of diffractogram means an amorphous halo. The authors at the same time claim that this is a response of the crystal structure. To clarify the authors' statement, it is necessary additionally add XRD diffractograms of crystalline PCL in the figure.

3. Table 1. The reaction scheme shown in Figure 1 corresponds only to the preparation of the PU–Si–â…¤ composition. It is necessary to pay attention to the text of the article and give appropriate explanations.

As a result, I will recommend the publication of this manuscript after major revision.  

Round 2

Reviewer 2 Report

Manuscript improved.